# Induction of Immune Tolerance in Islet Transplantation Using Apoptotic Donor Leukocytes

**DOI:** 10.3390/jcm10225306

**Published:** 2021-11-15

**Authors:** Naoya Sato, Shigeru Marubashi

**Affiliations:** Department of Hepato–Biliary–Pancreatic and Transplant Surgery, Fukushima Medical University, Hikagigaoka-1, Fukushima 960-1295, Japan; nawoya@fmu.ac.jp

**Keywords:** islet transplantation, tolerance induction, apoptotic donor lymphocyte

## Abstract

Allogeneic islet transplantation has become an effective treatment option for severe Type 1 diabetes with intractable impaired awareness due to hypoglycemic events. Although current immunosuppressive protocols effectively prevent the acute rejection associated with initial T cell activation in recipients, chronic rejection has remained an obstacle for achieving long-term allogeneic islet engraftment. The development of donor-specific immune tolerance to the allograft is the ultimate goal given its potential ability to overcome chronic rejection and disregard the need for maintenance immunosuppression, which may be toxic to islet grafts. Recently, a breakthrough in tolerance induction during allogeneic islet transplantation using apoptotic donor lymphocytes (ADLs) in a non-human primate model had been reported. Several studies have suggested that the clonal depletion, anergy, and expansion of the antigen-specific regulatory immune network are the mechanisms for donor-specific tolerance with ADLs, which act synergistically to induce robust transplant tolerance. This achievement represents a huge step forward toward the clinical application of immune tolerance induction. We herein summarize the reported operational induction therapies in islet transplantation using the ADLs. Moreover, a few obstacles for the engraftment of transplanted islets, such as islet immunogenicity and instant blood-mediated response, which need to be resolved in the future, are also discussed.

## 1. Introduction

Allogeneic pancreatic islet transplantation has been established as an effective option for severe Type 1 diabetes with intractable impaired awareness due to hypoglycemic events. Islet transplantation involves the intrahepatic delivery of donor-isolated islet cells that supplement insulin production, promoting the recovery of endogenous insulin secretion. Evidence has shown that the development of established immunosuppression protocol has improved outcomes of allogeneic islet transplantation [1]. However, the Collaborative Islet Transplant Registry data collected from islet transplant centers around the world reported 1- and 3-year insulin-independence rates of 71% and 24%, respectively [2]. A majority of islet transplant recipients return to some form of exogenous insulin usage within a few years of transplantation [3]. One factor associated with the long-term outcomes of transplanted islets is chronic rejection. Although current immunosuppression regimens effectively prevent acute rejection, which can suppress initial T cell priming by the donor antigen [4,5], no established immunosuppressive regimen has been effective in controlling chronic rejection.

When considering the adverse effects of immune suppression, some immunosuppressive drugs can be toxic to islet grafts [6], which could worsen the long-term functioning of the transplanted islets. Moreover, the long-term mediation of immunosuppressive drugs has also been associated with increased risk of infections [7], malignancies [8], cardiovascular disease [9,10], renal failure [11], de novo diabetes [12,13], and neurotoxicity [14]. With the increase in transplantation cases, a growing number of chronically suppressed transplant recipients struggle with such burdens.

The development of donor-specific immune tolerance to an allograft is the ultimate goal of any transplantation given its ability to possibly resolve chronic rejection and disregard the need for maintenance immunosuppression. Inducing donor-specific tolerance in animal transplantation models has been met with several challenges. Nonetheless, the accumulation of information has allowed the recent emergence of several key insights for operational tolerance induction, including the role of regulatory B cells (Bregs) for inducing or maintaining immunological tolerance.

As recently as 2019, Sigh et al. reported on a breakthrough in the tolerance induction protocol for allogeneic islet transplantation in non-human primate (NHP) models [15]. This approach for inducing donor-specific tolerance is unique in that it involves the strategic exposure of the recipient to donor antigens prior to transplantation. Several rodent models of allogeneic or xenogeneic transplantation have been evaluated on the impact of apoptotic donor lymphocyte infusion prior to transplantation on graft survival. The achievement by Sigh et al. represents a considerable step forward in the development of immune tolerance induction for human clinical applications.

We herein summarize the reported operational induction therapies in islet transplantation using the ADL protocol and review an essential mechanism of tolerance induction and maintenance based on the current knowledge gained from experimental animal models. A few obstacles hindering immunological tolerance and transplanted islet engraftment that need to be resolved in the future are also discussed.

## 2. General Understanding of the Relationship between the Immune System and Immunological Tolerance

The immune system can learn to discriminate between the self and nonself via a complex set of central and peripheral immune tolerance mechanisms. When considering immunological tolerance to the allograft, the high proportion of MHC alloreactive T cells, which generally range from 5 to 10% [16], has been considered as the main hindrance toward tolerance induction. Regarding T cell tolerance, central tolerance refers to the deletion of reactive clones within the thymus during negative selection. Peripheral T cell tolerance encompasses several mechanisms that take place outside the thymus, including peripheral deletion, anergy/exhaustion, and suppressive function of regulatory T cells (Tregs) [17]. The key lies in determining how to apply this mechanism of immunological tolerance, which is inherent in the body system, to induce donor-specific immune tolerance in transplantation.

During the immune destruction of the islet graft, the initial process of rejection is characterized by a rapid infiltration of innate immune cells to the grafts, which can be followed by an antigen-specific T cell response. The established immunosuppression protocol incorporating T cell depletion or anti-TNFa monoclonal antibodies could achieve T cell activation in allogeneic islet transplantation, which contributes to enhancing short-term graft survival [1]. However, the current protocol has been considered insufficient for controlling the humoral immune response, including antibody-mediated rejection, which is a significant mechanism of chronic allograft failure [18]. Indeed, B cells are known to mainly contribute to humoral immunity and boost cellular immunity. However, several experimental models have shown that B cell subsets ameliorate inflammation and autoimmunity disease, suggesting their capability for regulatory function, namely Bregs. Numerous bodies of evidence have indicated that B cells play essential roles in alloimmunity, including differentiation into antibody-producing plasma cells, sustaining long-term humoral immune memory, serving antigen-presenting cells (APCs), organizing the formation of tertiary lymphoid organs, and secreting pro- and anti-inflammatory cytokines (IL-10) [19]. Thus, therapeutic options targeting T cell–B cell interactions are of interest in the development of immunosuppressive protocols for transplantation [19]. Although a detailed description of B cell function in transplantation immunity is beyond the scope of the current paper, the following section summarizes exciting evidence obtained from islet transplantation animal models, demonstrating the significance of B cell functions in inducing and sustaining immunological tolerance.

## 3. Operative Tolerance Induction Using Low-Affinity TIM-1 mAb in an Islet Transplantation Model

TIM-1, which was initially reported to be a T cell costimulatory marker, is a member of the T cell immunoglobulin and mucin domain family of costimulatory molecules. However, an in vivo mice study by Ding et al. showed that TIM-1 was constitutively expressed on B cells rather than T cells and that 6–40% of TIM-1+ B cells express IL-10, including transitional, marginal zone, and follicular B cells [20]. Thus, reports have shown that TIM-1 is an inclusive marker of IL-10 + B cells. A mouse model of islet transplantation found that treatment with low-affinity anti-TIM-1, which has functional properties in the Breg development of recipients, leads to significantly prolonged islet allograft survival (approximately 30% of mice achieved long-term engraftment over 100 days) [20]. Interestingly, treating B cell-depleted recipients with anti-TIM-1 significantly enhanced IFN-g and ultimately prevented the commonly observed increase in Th2 cytokines. Therefore, B cells are required for anti-TIM-1-induced Th2 cytokine expression.

Moreover, a mouse islet allograft model found that anti-CD45RB antibody in combination with anti-TIM-1 antibody had a synergistic effect in inducing tolerance in all recipients. The dual antibody treatment significantly expanded regulatory B and T cells depending on the presence of recipient B cells with IL-10 activity [21]. Using B cell-deficient recipients or depleting B cells with anti-CD20 antibody abrogates the anti-CD45RB-induced tolerance following anti-TIM-1 dual antibody treatment. After exploring the reason why B cell depletion prevented the effects of dual antibody treatment on graft survival, the aforementioned study demonstrated that CD19 + CD5 + CD1d + B10 cells might play an important role only in early-stage transplantation tolerance induction following treatment. Additionally, the study also concluded that CD19 + TIM + B cells might play crucial roles in the whole process of tolerance induction and maintenance. These findings may explain why B cell depletion inhibited the effects of dual antibody treatment.

## 4. Important Evidence for Inducing Donor-Specific Tolerance to Preclinical Implementation

Although several promising approaches for tolerance induction in a rodent model of transplantation have been reported, as discussed earlier, very few have been translated to human or NHP transplantation models. Unlike laboratory mice, NHPs and humans already have a large collection of memory T and B cells at the time of transplantation. Heterologous immunity or the cross-reactive immune response has the potential to be alloreactive. For instance, up to 45% of anti-CMV T cell clones have been reported to be alloreactive [22]. Existing cross-reactive memory immune cells have been a significant hindrance to immunological tolerance in large animals or humans. Thus, tolerance induction in NHPs or humans would be more challenging than that in rodent models, partly due to cross-reactive memory T or B cells. Nevertheless, a few encouraging approaches have led us to believe that immunological tolerance can eventually be achieved in humans, such as mixed chimerism using hematopoietic cell transplantation [23,24] or ADL exposure [15].

One of the most advanced approaches for sustained tolerance is the use of hematopoietic cell transplantation to achieve durable chimerism, where the donor and recipient hematopoietic cells coexist at levels detectable by flow cytometry [24,25,26]. The central feature of the mixed chimerism-based transplant tolerance involves the intrathymic deletion of donor-reactive T cells by irradiation and/or anti-thymocyte globulin [27]. One study on kidney transplantation in NHPs reported that this protocol successfully promoted operationally sustained mixed chimerism in conditioning recipient and long-term engraftment [23]. Additionally, this approach has been successfully translated to both human leukocyte antigen-matched and -mismatched human renal transplant recipients with immunosuppression-free graft survival [24,28]. These results encourage attempts at extending this regimen to islet transplantation. Oura et al. reported the results of the mixed chimerism-based tolerance induction in islet transplantation in an NHP model, in which recipients were treated with a nonmyeloablative condition regimen that included total body irradiation, horse anti-thymocyte globulin anti-CD154 monoclonal antibody, and cyclosporine or anti-CD8 antibody (calcineurin inhibitor-free regimen) [29]. Accordingly, transient chimerism did not induce tolerance of islet graft survival, with islet function having been lost soon after the disappearance of chimerism [29]. Similar to islet transplantation, the induction of transient chimerism did not promote the tolerance of a heart allograft [30]. Thus, the effect of the mixed chimerism-based regimen for tolerance induction may be varied among the types of organs (success in kidney, failure in the islet and heart). Interestingly, Oura et al. also demonstrated that islet recipients had higher levels of serum inflammatory cytokines, such as TNFa and IL-17, compared to kidney recipients [29]. This finding suggests that high levels of inflammatory cytokines after islet transplantation could hinder the tolerance induction of islet grafts. As mentioned in another section, isolated islet grafts intrinsically provoke high levels of nonspecific systemic inflammation in recipients, which may need to be addressed in future studies to enhance graft survival and tolerance induction. The concept of immunological tolerance induction using apoptotic cells originates from research on autoimmune disease models. The strategy of inducing antigen-specific tolerance using ethylene carbodiimide (ECDI)-fixed splenocytes coupled with specific antigens or peptides has been considered as one of the most promising methods for the prevention and treatment of autoimmune diseases, including experimental autoimmune encephalomyelitis and autoimmune diabetes [31,32,33]. In these models, for instance, ECDI-fixed peripheral blood lymphocytes coupled with myelin basic protein (MBP) peptides selectively induced anergy in vitro in MBP-specific human helper T1 cells but not helper T2 cells, thereby preventing the onset of experimental autoimmune encephalomyelitis [34]. The ex vivo treatment of leukocytes with ECDI promoted rapid apoptosis after intravenous infusion [35]. As a normal function of various organisms, the clearance of apoptotic cells that occur under physiological conditions is a noninflammatory process [36]. During the process, apoptotic cells are quickly engulfed by macrophages for degradation, after which macrophages produce IL-10 in response to apoptosis cells [37]. Additionally, the uptake of apoptotic bodies by phagocytic cells can induce local TGF-b secretion, which promotes the generation and expansion of Tregs [38]. The immune suppression-inducing effects of apoptotic cells are supported by evidence suggesting that the apoptosis of peripheral blood lymphocytes induced by catecholamines, such as dopamine and dobutamine, could be associated with an immunosuppressive state in postoperative patients after cardiovascular surgery [39].

Based on the evidence of ECDI-fixed cell-based tolerance induction in autoimmune models, this approach has been applied to transplantation models, including islet transplantation. Studies have shown that pre- and post-transplant infusions of donor splenocytes treated with ECDI induce donor-specific tolerance and prolong graft survival in rodent organ transplantation models (kidney [40], cardiac [41], or skin [42]). The mechanisms by which donor ECDI-fixed splenocytes induce donor-specific tolerance in recipients have been found to be quite unique and interesting. Taba et al. demonstrated that the infused donor ECDI-fixed splenocytes target host allogeneic responses through various mechanisms, including clonal depletion, anergy, and the regulation of T cells with direct and indirect allospecificities, which act synergistically to induce robust transplant tolerance [43] (Figure A1). The aforementioned results demonstrated that infused donor ECDI-fixed splenocytes are rapidly internalized by recipient splenic marginal zone APCs, such as dendritic cells (DCs), macrophages, and B cells. Moreover, evidence has shown that apoptotic debris trigger IL-10 production [37] and that rapid and sustained IL-10 release from splenic marginal zone APCs occurs after their uptake of IV-infused, ECDI-treated, apoptotic leukocytes [35]. ECDI-fixed splenocytes activate and increase the number of Tregs and myeloid-derived suppressor cells [44].

Table 1 summarizes the available evidence regarding the effectiveness of ADL in inducing immunological tolerance in islet transplantation. Studies using rodent islet transplantation models have demonstrated the efficacy of the peritransplant infusion of ECDI-fixed donor splenocytes in not only allogeneic islet transplantation [15,45] but also xenogeneic islet transplantation [46,47]. Focusing on evidence that promotes clinical implementation, recent data regarding the NHP model reported by Singh et al. can be considered a breakthrough achievement [15]. Their protocol involves peritransplant infusions of MHC-DRB allele-matched apoptotic donor leukocytes under short-term immune suppressions, including antagonistic anti-CD40 antibody 2C10R4, rapamycin, soluble tumor necrosis factor receptor, and anti-interleukin 6 receptor antibody.

It is worth noting that the aforementioned study had been the first mechanistic study to demonstrate detailed immunological mechanisms associated with donor-specific tolerance in an NHP model. Several immunological monitoring assays, including Ki67+ proliferating cell tracking, alloreactive proliferation in mixed lymphocyte reaction, or T call immune repertoire profiling, have demonstrated that alloreactive effector T and B cells were depleted early after peritransplant ADL infusion. Notably, the mentioned study demonstrated the mechanisms by which this protocol-induced tolerance is characterized by notable immunological features, such as the generation of a regulatory network (myeloid-derived suppressor cells and Tr1, Tregs, NS, Breg, and B10 cells) and suppression of post-transplant expansion of alloreactive T cells in recipients (Figure A1). In particular, Tr1 cells, which are a crucial regulator for maintaining immune homeostasis in transplantation [48], were reported to be significantly prevalent within livers bearing islet grafts and draining lymph nodes in recipient monkeys.

Considering the successful results of the ADL-based tolerance induction protocol, the impact of one DRB-matched ADL infusion on the immune system of recipients warrants discussion. Interestingly, the frequency of circulating cells, such as Tregs, Tr1, and Bregs, which make up the regulatory network, is significantly higher in recipients with one-DRB-matched ADL than in those with fully mismatched ADLs [15]. Studies have shown that MHC-class II peptide complexes have a regulatory function on T cell activation, which is relevant to Treg suppression [49,50].

## 5. Effect of ADL on Tolerance Induction in Sensitized Recipients

Pre-existing donor-reactive memory T and B cells and donor-specific antibodies have been recognized as a cause of accelerated allograft rejection in sensitized allotransplantation [51,52]. Several challenges for tolerance induction with the ADL protocol in a sensitized model have been reported. Recently, Anil Dangi et al. demonstrated that pretransplant infusions of donor apoptotic cells in combination with anti-CD40L and rapamycin-induced significant prolongation of islet graft in allosensitized recipients (median survival time, 35 days) [53]. Although graft survival in sensitized recipients was enhanced, late graft rejection was not uncontrollable, which contradicted the results on the xenogeneic islet transplantation experiment in ECDI-treated recipients [46]. Interestingly, Anil Dangi et al. demonstrated that late graft rejection in recipients treated with triple therapy was associated with graft-infiltrating B and T cells and that additional B cell depletion was shown to be effective in preventing late rejection (islet survival of >180 days in ~80% of recipients). These results suggest that infiltrating B cells play a pivotal role in late islet rejection by promoting local T cell priming. Collectively, the infusion of donor ECDI-fixed splenocytes with the transient ISs in sensitized recipients effectively inhibited early alloreactive B cells, which may be reversed by concurrent B cell infiltration into the graft. Therefore, a strategy to control concurrent B cell infiltration is warranted in B cell-depleted recipients.

## 6. Obstacles toward Inducing Tolerance That May Be Specific to Islet Transplantation Immunogenicity of Islet Grafts

Islet grafts may be more likely to be resistant to immunological tolerance induction. As mentioned earlier, several studies on operational tolerance induction reported that the islet transplantation model failed the induction challenge, while the same induction protocol was successful in an organ transplantation model, such as cardiac grafts [54] or kidney grafts [29]. One possible factor hindering tolerance induction in islet transplantation might be the relatively higher immunogenicity of islet grafts compared to kidney grafts [29,55]. Given that pancreatic islets function as endocrine cells, islet grafts have relatively high cytokine secretion activity. Moreover, cell stresses during the isolation process promote islets inflammation, leading to an increase in the immunogenicity of the islet graft before transplantation. Our previous study on comprehensive gene expression in isolated islets prior to transplantation showed that proinflammatory gene clusters were dominantly upregulated in cultured islets, which significantly expressed the MCP-3 and GCP-2 genes [56]. The immunohistochemical staining of pancreatic islets in naïve pancreas showed that several chemokines, including MCP-3 and GCP-2, were expressed constitutively (unpublished data). These findings suggest that the stress of the isolation procedure partly triggered proinflammatory gene expression. Islet isolation involves multiple processes, namely the distention of the pancreas, digestion using collagenase, and purification. During each process, the islets should be damaged via hypoxia, warm ischemia, activated proteolytic enzymes released from acinar cells, mechanical stress, or oxidative stress [57,58]. Previous studies have shown that islet isolation induced proinflammatory cytokines and danger signals, including TNFa [59], monocyte chemoattractant protein-1 [60], and tissue factor [61]. These cytokines have been considered as significant barriers to islet engraftment.

Estimates have shown that roughly 50% of transplanted islets are irreversibly damaged during the peritransplant period, which usually spans from hours to days. Over 25% of transplanted islets are reported to be lost immediately after infusion into the portal vein [62]. A significant factor for immediate graft loss is the instant blood-mediated inflammatory reaction (IBMIR). This reaction is characterized by the activation of coagulation pathways, the release of proinflammatory cytokines, and the infiltration of innate immune cells [63], promoting the acute cell-mediated injury of the islets. Following islet transplantation, the release of proinflammatory cytokines and chemokines has been strongly associated with IBMIR. Although a direct link between IBMIR control and the development of alloimmune responses has yet to be established in allogeneic islet transplantation, further studies are warranted to develop protocols for controlling IBMIR. So far, it remains unknown as to whether ADL infusion could also regulate the IBMIR against the transplanted islets.

## 7. Preconditioning Islets Prior to Transplantation: A Possible Challenge for Reducing Immunogenicity of Islets

The short-term culture of isolated islets may precondition islet grafts for the improvement of islet transplantation outcomes. As described earlier, multiple proinflammatory cytokines are secreted from islets and act as strong stimulators of the host’s innate immune response. Evidence has shown that these molecules may serve as therapeutic targets for enhancing islet transplantation [64]. However, Citro et al. insisted that anti-inflammatory treatment targeting a single proinflammatory axis is insufficient given the redundancy and promiscuity of chemokine signaling mechanisms [65]. Our previous studies on the preconditioning treatment of islets with Mitomycin-C prior to transplantation showed that over half of the MMC-treated islets underwent engraftment without immunosuppression [66]. Additionally, suppressing the secretion of multiple chemoattractant cytokines from islets decreased the immunogenicity of isolated islets pretreated with MMC [67]. Consistent with our studies of islet pretreatment with MMC, the ex vivo pretreatment exposure of islets to sublethal genotoxic stressors, such as UV irradiation [68] or gamma irradiation [69], has been reported to reduce the post-grafting immune response and enhance islet engraftment. Collectively, the preconditioning of isolated islets with sublethal genotoxic stress can be a promising approach for reducing the immunogenicity of islet grafts and prolonging islet graft survival. It can be assumed that the preconditioning treatment for reducing the graft’s immunogenicity may have a synergistic effect on tolerance induction therapy, including ADL protocol.

Apart from suppressing the cytokine secretion of the islets, preconditioning of the islets may affect the status of tissue monocytes therein. Studies have revealed the significance of donor- and recipient-derived DCs in allograft rejection. Accordingly, donor DCs in islet tissues can migrate into the recipient’s secondary lymphatic tissue and circulation. In fact, Paolo et al. demonstrated the rapid migration of donor DCs to recipient lymphoid tissue as early as 3 h after allogeneic islet transplantation [70]. Moreover, compared to recipient DCs, the expression of maturity markers, including CD80, MHC class II, and CCR7, were increased in the donor DCs of 24 h-cultured islets, suggesting that donor DCs display much greater proliferative activity. Responder T cells recognize intact foreign MHC peptide complexes on the surface of donor DCs [71]. This pathway is characterized by the high frequency of responder T cells (~100-fold greater than that for a response to conventional protein Ag) and is critical for the initiation of alloresponse and acute graft rejection [72,73]. While mature donor DCs provoke a strong host immune response to islet grafts, one recent study showed that immature DCs derived from IL-10 treatment may effectively induce tolerance [74,75]. Indeed, much remains to be understood regarding whether an ex vivo preconditioning of the islet graft with MMC or irradiation could induce tolerogenic DCs and whether they are associated with the reduction in the alloresponse to treated islets.

## 8. Conclusions

The peritransplant infusion of ADLs under transit immune suppression is a robust protocol for inducing donor-specific tolerance in islet transplantation. This achievement encourages the belief that immunological tolerance can eventually be achieved in humans. Research must continue to address several unanswered questions, including the impact of ADL on sensitized recipients or recurrent alloimmunity as a cause of allogeneic islet failure in Type 1 diabetes or islet transplant recipients. Given the fact that tolerance induction seems to be more challenging in islet transplantation due to islet-specific factors, including the high immunogenicity of the islet itself and IBMIR, developing protocols for alleviating the systemic inflammatory response provoked by transplanted islets are warranted for the successful induction of immunological tolerance.

## Figures and Tables

**Table 1 jcm-10-05306-t001:** Summary of previous studies on tolerance induction using ECDI-donor splenocytes in islet transplantation animal experiments.

Summary of Previous Studies on Tolerance Induction Using ECDI-Donor Splenocytes in Islet Transplantation Animal Experiments					
Year	Authors	Tx Model	Induction Treatments	Tx Outcome	Mechanisms
2008	Luo et al.	mouse-to-mouse (allogeneicTx model)	ECDI-SPs	64% graft survival (>100 days)	Depletion of alloantigen-specific T cells CD4+CD25+ Tregs are required for tolerance induction by infusion of ECDI-treated donor splenocytes. PD-1/PD-L1 signaling pathway is associated with donor-specific tolerance induction by ECDI-fixed SPs
2013	Wang et al.	rat-to-mouse (xenogeneic Tx model)	ECDI-SPs	MST 48 days (18days for non-treated)	Anti-donor antibody: rat ECDI-SPs induced anti-rat IgGs (High levels of anti-rat IgG were detectable by day 14) C4d deposition: observed 14 days and 28 days in rat islet xenograft from recipients treated with rat ECDI-SPs. B cell activation: upregulated expression of costimulatory molecules CD80, CD86, CD40, and OX40L B cell infiltration: observed in ECDI-SPs recipients 2 or 4 weeks after Tx.
			ECDI-SPs with B cell depletion	100% graft survival (>100 days)	Anti-donor antibody: no production of anti-rat antibodies of all IgG subclasses at 14 days after ECDI-SPs C4d deposition: negative B cell activation: N.A B cell infiltration: minimal infiltration of B220 cells xeno-specific T-cell priming: suppressed memory T cell generation: suppressed rebound B cells: xeno-donor-specific B cell unresponsiveness
2017	Kang et al.	pig-to-mouse (xenogeneic Tx model)	no treatment	acute rejection (by day 7–26 post-transplantation)	B cell infiltration to grafts; prominent (cf. minimal infiltration of B cells to graft in alloislet Tx) High expression of IL-17 on CD4 and CD8T cell from rejected mice (cf. high levels of IFN-r on T cell from rejected mice in alloislet Tx.)
			ECDI-SPs only	no prolongation of graft survival	N.A
			ECDI-SPs with B cell depletion	prolongation of graft survival(40% graft >100 days)	N.A
			ECDI-SPs with B cell depletion and transient rapamycin	1. prolongation of graft survival(65% graft >100 days) 2. late rejection between day 100 and 200 post-transplantation (B cell reconstitution)	initial phase(day 21–70 post transplantation) anti-pig IL-17 response: suppressed rejection phase B cell infiltration to graft: aggressive infiltration of B cells to graft anti-pig antibody production: minimal anti-pig INFr response; observed in indirect donor stimulation, but not direct donor stimulation
2019	A.Sigh	HNP-to-NHP (MHC lassⅡ matched) (allogeneic Tx model)	ECDI-SPs and transient ISs (anti-CD40, anti-IL-6R, anti-TNFaR, rapamycin)	long-term tolerance (100%)	antigen-specific regulatory networks: Tr1, Breg, B10, MDSC One DRB-matched ECDI-SPs; expanded alloantigen-specific regulation
2020	Dangi et al.	mouse(B6)-to-mouse(Balb/c) (allogeneic islet transplantation)	ECDI-SPs only	no prolongation of graft survival	
			ECDI-SPs and transient ISs (anti-CD40 and rapamycin)	MST 35 days	donor-specific graft-infiltrating T cells; inhibited expansion of donor-specific memory B cells; inhibited infiltrating B cells in late rejected islets with high expression of CD40 and CD86
			ECDI-SPs and transient ISs (anti-CD40 and rapamycin)B cell depletion	islet survival of >180 days in ~80% of recipients	

## Data Availability

Not applicable.

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
