# Peer review of "Induction of Immune Tolerance in Islet Transplantation Using Apoptotic Donor Leukocytes"

_jcm, 2021, doi:10.3390/jcm10225306_

Round 1

Reviewer 1 Report

The development of donor-specific immune tolerance to a graft would solve the issues with chronic rejection and the need for life-long immunosuppression. Sato and Marubashi aim to provide an overview of donor-specific immune tolerance methods for islet transplantation.

  • The title of section 4. is ‘Important evidence for inducing donor-specific tolerance to preclinical implementation’. However, the evidence is described in sections 5. and 6. Either the title is misleading and should be changed, or 5. and 6. become part of section 4.

  • The authors conclude that the effect of the mixed chimerism-based therapy is organ-specific (success in kidney, failure in heart/islets). Can you make such a solid conclusion based on one study?

  • In section 6. the authors refer to Amar et al., but this reference is not mentioned in the reference list.

  • Islet recipients are often infused with islets of 3-4 donors, will this not be an obstacle for ADL infusion, since then also ADL of multiple donors are needed? This should be discussed.

  • Section 9. about the IBMIR does not really seem to fit the rest of the story. What is the reason behind this paragraph? Or should it be part of section 8.?

  • The manuscript is giving mixed signals about the focus of the review. Based on the title and the last paragraph of the introduction, you expect multiple tolerance-inducing therapies to be discussed. However, it is mainly ADL infusion that is discussed and in the conclusion the authors only refer to this therapy. The discussion of other therapies is limited, the only other therapy that is shortly discussed is the mixed chimerism-based therapy. But there are more treatment options investigated to induce tolerance, for example with MSCs (Kenyon et al. 2021 American Journal of Transplantation). The authors should choose to either expand the discussion of different methods or more clearly state that their focus is on ADL infusion.

  • The manuscript would benefit from grammatical editing.

Author Response

【Reviewer 1】

We take this opportunity to express our gratitude to the reviewer for constructive and useful remarks. Their comments allowed us to identify areas in our manuscript that needed modification and clarification. Based on the comments from the reviewers, we revised the manuscript carefully in a manner of point-by-point response to the comments. The following text is our responses to the comments from the reviewer. Each comment from the reviewer is given in the underlined text, followed by our response.

  1. The title of section 4. is ‘Important evidence for inducing donor-specific tolerance to preclinical implementation’. However, the evidence is described in sections 5. and 6. Either the title is misleading and should be changed, or 5. and 6. become part of section 4.

【our response】

Thank you for the suggestive comment to this point. According to your comment, the sections entitled 5 and 6 were included in section 4.

  1. The authors conclude that the effect of the mixed chimerism-based therapy is organ-specific (success in kidney, failure in heart/islets). Can you make such a solid conclusion based on one study?

【our response】

Thank you for this question clarifying the issue that may be misleading. I agree with you that such a solid conclusion can’t be derived. We made a change in the description as below.

The revised sentence

Thus, the effect of the mixed chimerism-based regimen for tolerance induction may be varied between the type of organs (success in kidney, failure in the islet and heart).

  1. In section 6. the authors refer to Amar et al., but this reference is not mentioned in the reference list.

【our response】

Thank you for pointing out this mistake. We added the reference into the appropriate sentence.

  1. Islet recipients are often infused with islets of 3-4 donors, will this not be an obstacle for ADL infusion, since then also ADL of multiple donors are needed? This should be discussed.

【Our response】

Thank you for the pivotal question. I understand your comment is a natural one if multiple infusion of ADL is needed. However, according to the evidence showing the efficacy of ADL to tolerance induction in the NHP model (reference No.15), isolated islets from one donor NHP is enough for normalizing blood sugar levels in the one recipient. It remains unknown whether the result would apply to human islet transplantation. However, I don’t believe that researchers who are pushing the ADL study to clinical study are assuming multiple infusions of islet graft along with the ADL for achieving insulin independence. Therefore, we didn’t add any description about this point.

  1. Section 9. about the IBMIR does not really seem to fit the rest of the story. What is the reason behind this paragraph? Or should it be part of section 8.?

【our response】Thank you for this suggestion. according to your suggestion, we changed that section 9 was included in section 8 

  1. The manuscript is giving mixed signals about the focus of the review. Based on the title and the last paragraph of the introduction, you expect multiple tolerance-inducing therapies to be discussed. However, it is mainly ADL infusion that is discussed and in the conclusion the authors only refer to this therapy. The discussion of other therapies is limited, the only other therapy that is shortly discussed is the mixed chimerism-based therapy. But there are more treatment options investigated to induce tolerance, for example with MSCs (Kenyon et al. 2021 American Journal of Transplantation). The authors should choose to either expand the discussion of different methods or more clearly state that their focus is on ADL infusion.

【Our response】

Thank you for pointing out the critical issue in the manuscript. We totally agree with your suggestion. A message that we want to tell readers of this review is all about ADL protocol for tolerance induction in islet transplantation because the efficacy has been proved in NHP allogeneic islet transplantation (in the preclinical stage). Therefore, we made some changes in the whole manuscript so that our intentions or focus could be expressed.

  1. The manuscript would benefit from grammatical editing.

Let me tell you that we have submitted the 1st manuscript after the English edition by the Editage company. However, we found that some mistakes in grammar as well as spelling mistakes. Unfortunately, due to the limited time to deadline, we couldn’t have much time for new grammatical editing for the revised manuscript. Could you give us a chance to edit this manuscript by another experienced English editor?

Reviewer 2 Report

The authors did a rigorous literature review on the selected topic. The text is generally well written and comprehensive to the readers. The reviewer have few minor suggestions that could improve the manuscript.

  • The author may include a short separate section on non-immune or stromal cell mediated tolerance. Previous studies show cells like MSCs, HSC and Sertoli cells play an important role on tolerance during islet transplantation.
  • The manuscript could also benefit if author briefly comment on recent advanced technologies that are available to cause immune tolerance. eg, investigational technologies like micro and macro encapsulation of islets, CRISPR-Cas9 mediated gene-editing in xenotransplantation, stem cell derived beta cell islet replacements, etc.,

The author used sufficient references.

Minor language checks are needed.

Author Response

We take this opportunity to express our gratitude to the reviewer for constructive and useful remarks. Their comments allowed us to identify areas in our manuscript that needed modification and clarification. Based on the comments from the reviewers, we revised the manuscript carefully in a manner of point-by-point response to the comments. The following text is our responses to the comments from the reviewer. Each comment from the reviewer is given in the underlined text, followed by our response.

  1. The author may include a short separate section on non-immune or stromal cell mediated tolerance. Previous studies show cells like MSCs, HSC and Sertoli cells play an important role on tolerance during islet transplantation. The manuscript could also benefit if author briefly comment on recent advanced technologies that are available to cause immune tolerance. eg, investigational technologies like micro and macro encapsulation of islets, CRISPR-Cas9 mediated gene-editing in xenotransplantation, stem cell derived beta cell islet replacements, etc.,

【Our response】

Thank you for the suggestive and informative comments. Actually, we considered adding a few sections for recent advances in tolerance induction, including stromal cell or MSCs, HSC, or encapsulation technique. However, as suggested by reviewer 1, the manuscript should be focused on ADL protocol. We take this suggestion seriously because the figure and table1 that we made in the manuscript are only focused on the ADL. Therefore, we made some changes in the whole manuscript so that our intentions or focus could be expressed.

  1. The author used sufficient references.

【our response】Thank you for pointing out this mistake. We added a reference into the appropriate sentence in section 6.

  1. Minor language checks are needed.

【our response】

Let me tell you that we have submitted the 1st manuscript after the English edition by the Editage company. However, we found that some mistakes in grammar as well as spelling mistakes. Unfortunately, due to the limited time to deadline, we couldn’t have much time for new grammatical editing for the revised manuscript. Could you give us a chance to edit this manuscript by another experienced English editor?

Round 2

Reviewer 1 Report

The changes made by the authors are sufficient, the focus of the manuscript is now more clear and adequately referenced. As the authors suggest, I would recommend to give them some more time to let the manuscript be edited by a native/experienced English speaker.